# A New Efficient method for eliminating minor components

# Conference Submissions

## Abstract

We present a new gradient method for quadratic programming, named EM(Eliminating Minor Component Values), which eliminate the component in the direction of large eigenvalues when the current position is in the direction of small eigenvalues, thereby reducing the increase in the component of large eigenvalues in the next iteration. Numerical experiments show that the EM method has significant advantages over some commonly used optimization methods, such as BB and CBB.

## 1 Introduction

In this paper, we present a gradient updating method for solving the unconstrained optimization problem ,the purpose is to minimize an objective function

$$minf(x) = \frac{1}{2}x^T A x - b^T x \tag{1}$$

where $x \in \mathbb{R}^n, b \in \mathbb{R}^n, A \in \mathbb{R}^{n \times n}$ is a symmetric and positive definite matrix. The common solution methods for solving Eq.(1) are iterarive methods of the following form

$$x_{k+1} = x_k - \alpha_k \nabla f(x_k) \tag{2}$$

where $\alpha_k$ is a steplenth,gradient descent method and its variants are the most common optimization method.if we minimizes Eq.(1) with exact line search,then we get

$$\alpha_k = \frac{\nabla f_k^T \nabla f_k}{\nabla f_k^T A \nabla f_k} = \frac{g_k^T g_k}{g_k^T A g_k} \tag{3}$$

$$r_k = \frac{1}{2\alpha_k} = \frac{g_k^T A g_k}{2 g_k^T g_k} \tag{4}$$

this method proposed by A.Cauchy (1847) is called steepest descent method ,so $\alpha_k^{SD}$ is also called Cauchy step length. the method's convergence rate is very sensitive to ill condition number and may be very slow ,when the f(x) is quadratic $x_k$ will satisfy the

$$\frac{f(x_{k+1}) - f(x^*)}{f(x_k) - f(x^*)} \leq (\frac{\lambda_1 - \lambda_n}{\lambda_1 + \lambda_n})^2 \tag{5}$$

During the iteration process, the SD method exhibits a zigzag phenomena which was explained by Akaike (1959), J.BARZILAI & J.M.BORWEIN (1988) presented a nonmonotone steplength which certain quisi-Newton method, it has two choice for $a_k$,respectively:

$$\alpha_k^{BB1} = \frac{s_{k-1}^T s_{k-1}}{s_{k-1}^T y_{k-1}} \tag{6}$$

$$\alpha_k^{BB2} = \frac{s_{k-1}^T y_{k-1}}{y_{k-1}^T y_{k-1}} \tag{7}$$

where $s_{k-1} = x_k - x_{k-1}$ and $y_{k-1} = g_k - g_{k-1}$ The BB step can be seen Cauchy step with previous iteration. Barzilai and Borwein proved R-superlinear convergence rate in two dimension. for general n dimensional convex quadratic case, the method is convergent too and has a properties of R-linear rate of convergence(Yuan (2008) ). there are some optimization methods based on gradient, YH (2003) decrease the gradient norm , Yuan (2006) and YH (2005) design a alternate steps , in two dimension case, it could convergence 3 steps. Asmundis;Serafino (2013) propose SDA with a fixed stepsiz in sussesive steps. and SDC (R.De Asmundisdi SerafinoD (2014)) adding Cauchy step comparing SDA. Sun C (2020) propose new step size based on Cauchy stepsize. Kalousek(Z (2015)) select random stepsize at some range.Raydan M (2002) propose a relax method .Raydan M (2002) introduce RSD which accelerates convergence by introducing a relaxation parameter between 0 and 2 in the standard Cauchy method, The CBB method is a nonmonotone approach that selects two directions to synthesize a new direction, they also propose CBB method which is a combination of the SD and BB method ,CBB method is essentially equivalent to the steepest descent method over two consecutive steps. We conducted research on the reciprocal of the optimal step sizes of the SD and CBB methods for $r$ (where $r$ can be seen as the gradient value projected onto each eigenvalue component), We propose a method based on $r$, which involves selecting multiple fixed values of $r$ at positions with smaller $r$ values during the iteration process. This approach aims to reduce the increase in the direction of larger eigenvalues in the next iteration.

## 2   R ANYLSIS

Assuming the initial value is $x_0$, we use the SD method for updating

$$x_0^s = x_0 - \alpha_0 g_0 \tag{8}$$

than we search in the $Ag_0$ direction and find the point $x_0^A$, the vecotrs $\overrightarrow{x_0 x_0^s}$ and $\overrightarrow{x_0^s x_0^A}$ are perpendicular. than we find the symmetric point $x_1$ of $x_0^A$ with respect to $x_0^s$.it is obvious $|x_1 x_0^s| = |x_0^s x_0^A|$,as shown in Figure(1). In order to make the analysis more convenient and intuitive,considering a situation the objective function is a simple $n$ dimensions hyper-ellipsoid stimulating Eq.(1)

$$f(x) = \sum_{i=1}^{n} a^{(i)} x^{(i)^2} \tag{9}$$

$$r = \frac{\sum_{i=1}^{n} a^{(i)^3} x^{(i)^2}}{\sum_{i=1}^{n} a^{(i)^2} x^{(i)^2}} = \frac{\sum_{i=1}^{n} a^{(i)} g^{(i)^2}}{\sum_{i=1}^{n} g^{(i)^2}} \tag{10}$$

where $0 < a^{(n)} \le a^{(n-1)} \le ...... \le a^{(1)}$,$a^{(1)} \gg a^{(n)}$,$g^{(i)} = 2a^{(i)} x^{(i)}$

$$x_0^s = x_0 - \frac{\nabla f(x_0)}{2r_0} \tag{11}$$

$$x_0^{s(i)} = x_0^{(i)}(1 - \frac{a^{(i)}}{r_0}) = x_0^{(i)} \mu_0^{(i)} \tag{12}$$

we define $v_0 = Ag_0$ , $l_0 = \|x_0 x_0^s\|$ , $l_0^A = \|x_0 x_0^A\|$, $\theta_0$ is the angle between $g_0$ and $v_0$ we have

$$cos\theta_0 = \frac{g_0^T v_0}{\|g_0\|\|v_0\|} = r_0 [\frac{\sum_{i=1}^{n} a^{(i)^2} x_0^{(i)^2}}{\sum_{i=1}^{n} a^{(i)^4} x_0^{(i)^2}}]^{\frac{1}{2}} \tag{13}$$

$$\frac{\|v_0\|}{\|g_0\|} = 2[\frac{\sum_{i=1}^{n} a^{(i)^2} x_0^{(i)^2}}{\sum_{i=1}^{n} a^{(i)^4} x_0^{(i)^2}}]^{\frac{1}{2}} \tag{14}$$

$$l_0^A = l_0 / cos\theta_0 = \frac{(\sum_{i=1}^{n} a^{(i)^4} x_0^{(i)^2})^{\frac{1}{2}}}{r_0^2} \tag{15}$$

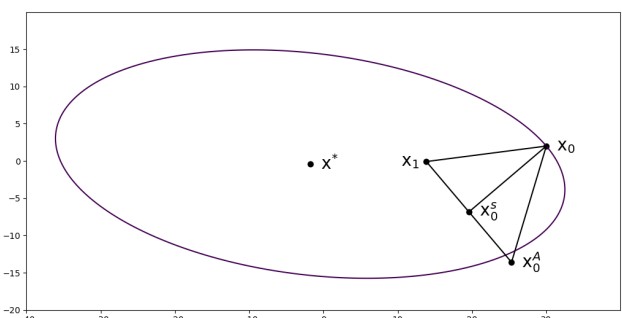

Figure 1: SD and CBB

we have $x_0^{A(i)} = x_0^{(i)}[1 - (\frac{a^{(i)}}{r_0})^2]$ because $x_1$ is a symmetric points of $x_0^A$ about $x_0^s$, so $x_1 = 2x_0^s - x_0^A$, $x_1^{(i)} = x_0^{(i)}(1 - \frac{a^{(i)}}{r_0})^2 = x_0^{(i)}\mu_0^{(i)^2}$ $g_1^{(i)} = g_0^{(i)}(1 - \frac{a^{(i)}}{r_0})^2 = g_0^{(i)}\mu_0^{(i)^2}$

It is evident that $x_1$ is the result obtained after applying the CBB method which is equivalent to using SD method with the same steplenth in two consecutive iterations. From the above analysis and Figure(1), we can see that the CBB update direction is symmetric to the $Ag$ direction with the current gradient as the axis.

## 3 FURTHER DISCUSSION

We will conduct a more in-depth investigation of the value of $r$ based on the SD and CBB methods.

### 3.1 TWO DIMENSIONS CASE

For SD method,from Eq.(10) we have $r_0 = \frac{a^{(1)}g_0^{(1)^2} + a^{(2)}g_0^{(2)^2}}{g_0^{(1)^2} + g_0^{(2)^2}}$,

$$r_1 = \frac{a^{(1)}g_0^{(1)^2}(r_0 - a^{(1)})^2 + a^{(2)}g_0^{(2)^2}(r_0 - a^{(2)})^2}{g_0^{(1)^2}(r_0 - a^{(1)})^2 + g_0^{(2)^2}(r_0 - a^{(2)})^2} = \frac{a^{(2)}g_0^{(1)^2} + a^{(1)}g_0^{(2)^2}}{g_0^{(1)^2} + g_0^{(2)^2}} \tag{16}$$

we have $r_2 = r_0$, $r_0 + r_1 = r_1 + r_2 = r_k + r_{k+1} = a^{(1)} + a^{(2)}$,then we define $e^{(i)} = (1 - \frac{a^{(i)}}{r_u})(1 - \frac{a^{(i)}}{r_d})$,then $e^{(1)} = e^{(2)} = e$ in two dimensions, $r$ will immediately achieve stable state , so $x_{2k}^{(i)} = x_0^{(i)}e^k$,$g_{2k}^{(i)} = g_0^{(i)}e^k$,$g^{(1)}$ and $g^{(2)}$ have the same decrease rate every two steps.

$$e = \frac{(a^{(1)} - a^{(2)})^2 g_0^{(1)^2} g_0^{(2)^2}}{a^{(1)}a^{(2)}(g_0^{(1)^4} + g_0^{(2)^4}) + (a^{(1)^2} + a^{(2)^2})g_0^{(1)^2}g_0^{(2)^2}} = \frac{(\nu - 1)^2}{(1 + \nu)^2 + \nu(\mu - \frac{1}{\mu})^2} \tag{17}$$

where $\nu = \frac{a^{(1)}}{a^{(2)}}$,$\mu = \frac{g_0^{(1)}}{g_0^{(2)}}$,when $\mu = 1$($\nu$ was set as fixed value), the $e$ reaches the maximum. so $e = \frac{(\nu - 1)^2}{(\nu + 1)^2}$, $r = \frac{a^{(1)} + a^{(2)}}{2}$,this is the worst case, $e$ is SD convergence speed and very close to 1.the convergenc speed increase with the inreasing $\mu$, so $r = a^{(2)}\frac{\nu\mu^2 + 1}{\mu^2 + 1}$

$r$ and $a^{(1)}$ are coming close. the closer the r was to $a^{(1)}$, the smaller the $e$ and the faster the convergence. $r$ decide the convergence speed, we can accelerate the convergence speed by adjustments to $r$ value.

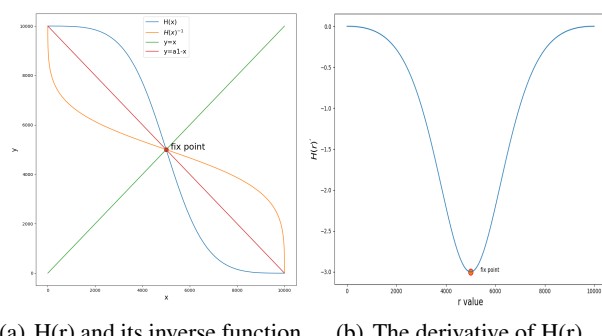

(a) H(r) and its inverse function    (b) The derivative of H(r)

Figure 2: CBB function,$a_1 = 10000, a_2 = 0.1$

For the CBB method, At the k-th iteration, if we treat $r_k$ as a continuous variable $r$,and regard $r_{k+1}$ as a function $H(r)$ of variable $r$ ,then

$$H(r) = \frac{a^{(1)}(r - a^{(1)})^3 - a^{(2)}(r - a^{(2)})^3}{(r - a^{(1)})^3 - (r - a^{(2)})^3} \tag{18}$$

By finding the fixed points of $H(r)$,we can obtain the values $r_{fix} = \frac{a^{(1)}+a^{(2)}}{2}$. the derivative at the fixed point is $H(r_{fix})^{'} = -3$,therefore, it can be concluded that the fix point $r_{fix}$ is a repulsive point. Meanwhile, the derivative value at the fixed point is a constant value of $-3$,which does not change with $a^{(1)}$ and $a^{(1)}$. From Figure(2a), it can be observed that when $r$ is small, the value of $H(r)$is close to $a^{(1)}$. Similarly, when $r$ is large,the value of $H(r)$ is close to $a^{(2)}$,$H(r)$oscillates rapidly towards $a^{(1)}$ and $a^{(2)}$. When the value of $r$ is near the fixed point, $H(r)$oscillates more slowly towards $a^{(1)}$ and $a^{(2)}$. From Figure(2b),Because the derivative of the $H(r)$ function has a large value at the equilibrium point, the change in $r$ is very rapid. So in the two-dimensional case, the value of $r$ using the CBB method will quickly approach the two characteristic values. If the current value of $r$ is a characteristic value, the component in the direction of that characteristic value will be eliminated after the next iteration, until the stopping condition is met.

### 3.2   N DIMENSIONS CASE

Acording to Nocedal J. (2022) and Asmundis;Serafino (2013) research, sum of two consecutive the reciprocal of steps approach to the asymptotic value of toting up the maximum and minimum eigenvalue in SD method(Forsythe (1968)) we will analyzed it from the perspective of $r$.

$$r^{SD}_{k+1} = \frac{\sum_{i=1}^n a^{(i)} {g^{(i)}_{k+1}}^2}{\sum_{i=1}^n {g^{(i)}_{k+1}}^2} = \frac{\sum_{i=1}^n a^{(i)} {g^{(i)}_k}^2 {\mu^{(i)}_k}^2}{\sum_{i=1}^n {g^{(i)}_k}^2 {\mu^{(i)}_k}^2} = \frac{\sum_{i=1}^n a^{(i)} {g^{(i)}_k}^2 W^{(i)}_k}{\sum_{i=1}^n {g^{(i)}_k}^2 W^{(i)}_k} \tag{19}$$

where $\mu^{(i)}_k = 1 - \frac{a^{(i)}}{r_k}$,$W^{(i)}_k = (r_k - a^{(i)})^2$

$$r^{SD}_k + r^{SD}_{k+1} = \frac{\sum_{i=1}^n \sum_{j=1}^n {g^{(i)}_k}^2 {g^{(j)}_k}^2 A(a^{(i)}, a^{(j)})}{\sum_{i=1}^n \sum_{j=1}^n {g^{(i)}_k}^2 {g^{(j)}_k}^2 B(a^{(i)}, a^{(j)})} \tag{20}$$

then we can see $A(a^{(i)}, a^{(j)})$ and $B(a^{(i)}, a^{(j)})$ as the different weight of the numerator and denominator of Eq(20).so the bigger the difference between the $a^{(i)}$and $a^{(j)}$, the greater the weight in $a^{(i)}$and $a^{(j)}$. From Figure(3), $x$ and $y$ corresponding to larger values of $A$ and $B$ more center at the top left corner area and the bottom right corner area. the $x$ and $y$ in other areas lead to smaller value of $A$ and $B$,so only the $a^{(i)}$ and $a^{(j)}$ locate in the maximum eigenvector direction area apporximate $a^{(1)}$ and the minimum eigenvector direciton area approximate $a^{(n)}$ have the biggest weight.So

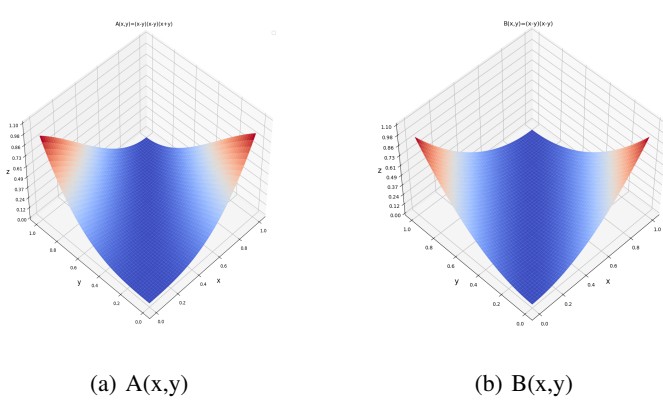

(a) A(x,y)         (b) B(x,y)

Figure 3: A(x,y) and B(x,y), $0.01 \leq x \leq 1, 0.01 \leq y \leq 1$

Eq(20) is mainly affected by the value at maximum eigenvalue ares and minimum eigenvalue area. after a few step, the system will fall into a state of balance situation,so $r_k + r_{k+1} \approx a^{(1)} + a^{(n)}$, Similar to the 2-dimensional case, in an $n$ dimensional scenario, the system will eventually settle into an equilibrium state, oscillating back and forth between two values of $r$.

for CBB methods, each iteration can be viewd as two consecutive SD iterations. so we have

$$r_{k+1}^{CBB} = \frac{\sum_{i=1}^{n} a^{(i)} g_{k+1}^{(i)^2}}{\sum_{i=1}^{n} g_{k+1}^{(i)^2}} = \frac{\sum_{i=1}^{n} a^{(i)} g_k^{(i)^2} \mu_k^{(i)^4}}{\sum_{i=1}^{n} g_k^{(i)^2} \mu_k^{(i)^4}} = \frac{\sum_{i=1}^{n} a^{(i)} g_k^{(i)^2} W_k^{(i)^2}}{\sum_{i=1}^{n} g_k^{(i)^2} W_k^{(i)^2}} \tag{21}$$

Let's analyze the values of $r$ before and after consecutive iterations. Unlike the 2-dimensional case, $H(r)$ is not a deterministic function of $r$ in n-dimensional space.However, we can observe the relationship between consecutive $r$ values during the process of the CBB method.From Figure 4(b), it can be seen that $H(r)$ almost covers the entire space. However, it has a main trajectory similar to the two-dimensional case. Meanwhile, most of the points are located in the upper-left and lower-right corners. This implies that if the current value of $r$ is small or large, the value of $r$ after the next iteration will become large or small, respectively. If the value of $r_k$ is large, the weight corresponding to the larger eigenvalues will relatively decrease, while the weight corresponding to the smaller eigenvalues will relatively increase, causing the value of $r_{k+1}$ to shift towards the direction of smaller eigenvalues.If the value of $r_k$ is small, the weight corresponding to the smaller eigenvalues will relatively decrease, while the weight corresponding to the larger eigenvalues will relatively increase, causing the value of $r_{k+1}$ to shift towards the direction of larger eigenvalues. As shown in the Figure(5), just like the previous analysis, the r value of the SD method oscillates between two relatively determined values of size, while the r value of the CBB method alternates between large and small values. As shown in the Figure(6) , The distribution of r values for the SD method are concentrated at two fixed values, whereas the CBB method has a distribution across all values, with a higher concentration in the directions of the maximum and minimum eigenvalues and a lower concentration in the directions of intermediate eigenvalues.

From Eqs.(19) and (21) , We can regard $\mu^{(i)}$ as the weights of different gradients. $r$ value will seesaw between larger eigenvalue area and smaller eigenvalue area generally.the component of small eigenvalue determine the convergence rate and is hard to reduce .for SD method,$r$ will stabilize in two certain value which means to be relatively fixed decrase rate. Comparing the SD method ,the CBB method's $r$ value have more wider range change , and have higher descent rate in the direction of small eigenvalue also.

## 4  OUR METHOD

According to the previous analysis , if the current $r$ value is very small, there will be a substantial increase in the direction of the larger eigenvalue during the next iteration. Assuming that the condition number $k$ is a value much greater than 1 . If the $r$ value is close to the smallest eigenvalue

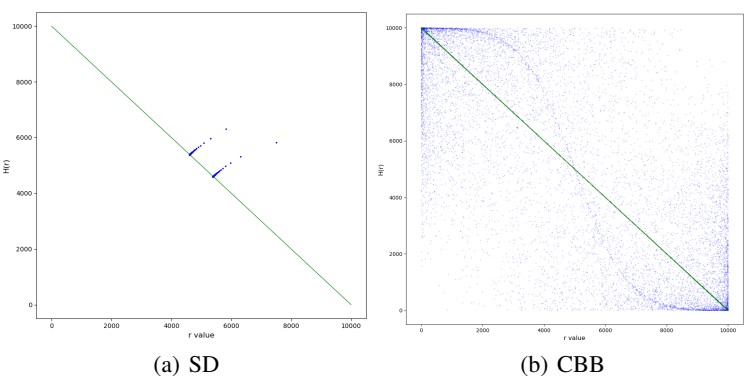

(a) SD        (b) CBB

Figure 4: SD and CBB H(r) value

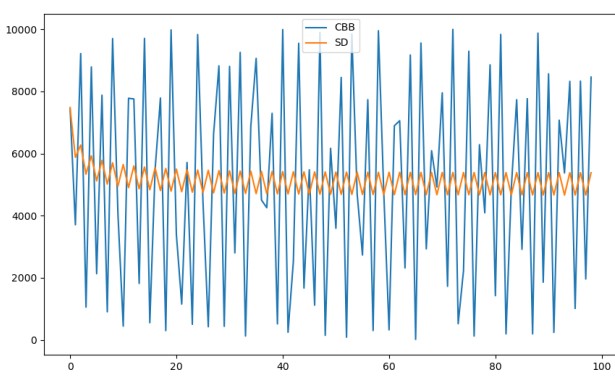

Figure 5: SD and CBB r value

$a^{(n)}$, according to the SD method, the value after the next iteration in the direction of the largest eigenvalue $a^{(1)}$ will be $k$ times its previous value, whereas for the CBB method, it will be $k^2$ times, Simultaneously, all components will experience substantial growth, even though the original values of these components are not large. If the current $r$ value is small, as can be deduced from its calculation formula, the components outside the direction of the small eigenvalues will also be extremely small. We refer to these smaller components as spikes. If these spikes can be effectively eliminated prior to the next iteration, it would be beneficial to the overall computation process.

We refer to the interval from $a^{(n)}$ and $\frac{1}{2}(a^{(1)} + a^{(n)})$ as the lower half-interval, and the interval from $\frac{1}{2}(a^{(1)} + a^{(n)})$ to $a^{(1)}$ as the upper half-interval.Assuming that $a^{(1)}$ is much larger than $a^n$ and

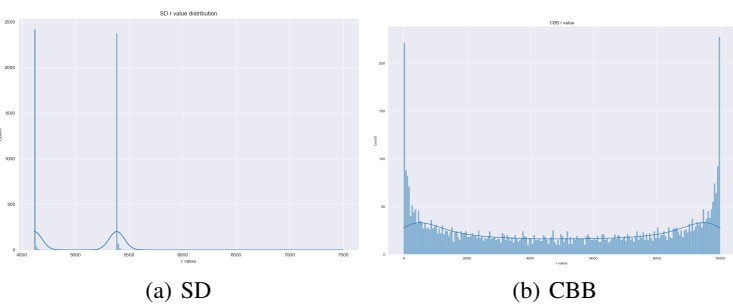

(a) SD        (b) CBB

Figure 6: SD and CBB r value distribution

$a^{(n)}$ is close to $0$ , the lower half-interval can be approximated as from $0$ to $\frac{1}{2}a^{(1)}$, and the upper half-interval can be approximated as from $\frac{1}{2}a^{(1)}$ to $a^{(1)}$.

When the $r$ falls into the upper half-interval, the growth rates in all eigenvalue directions are less than 1, with those in the direction of the smaller eigenvalues being close to 1. However, when the $r$ falls into the lower half-interval, the growth rates may exceed 1, potentially significantly so in the direction of the larger eigenvalues.

To eliminate these spikes, we apply different treatments to two distinct intervals.for the upper half-interval interval, we selectively choose some $r$ values to reduce the components in the direction of the larger eigenvalues. we evenly space out $m$ values, and it can be understood that iterating $m$ times is equivalent to a polynomial of degree $m$.

In our case, We divide the space into 5 equal parts, which results in 6 values, namely $a^{(1)}$, $0.9a^{(1)}$,$0.8a^{(1)}$, $0.7a^{(1)}$, $0.6a^{(1)}$, and $0.5a^{(1)}$, the corresponding descending function is

$$\begin{aligned} g(x) =&(1 - \frac{x}{a^{(1)}})(1 - \frac{x}{0.9a^{(1)}})(1 - \frac{x}{0.8a^{(1)}})(1 - \frac{x}{0.7a^{(1)}})(1 - \frac{x}{0.6a^{(1)}})(1 - \frac{x}{0.5a^{(1)}}) \\ =& 0.1512(1 - y)(0.9 - y)(0.8 - y)(0.7 - y)(0.6 - y)(0.5 - y) \end{aligned} \tag{22}$$

where $y = \frac{x}{a^{(1)}}$, within this interval, the maximum absolute value of $g(x)$ after iterating 6 times is approximately 0.000142128,If we use the CBB method, it is equivalent to squaring $g(x)$, resulting in a maximum value of 2.0200368384e-8 , this indicate a significant decline.

For the lower half-interval, we employ a strategy that ensures both a decrease in the components along the smaller eigenvalues and prevents the substantial growth in the direction of the larger eigenvalues mentioned earlier during the next iteration. this needs to satisfy the following condition $|(1 - \frac{x}{r})(1 - \frac{x}{s})| < 1$ , Since two consecutive iterations correspond to a quadratic equation, the absolute value of the parabolic vertex and the maximum eigenvalue direction should be less than 1. Therefore, we set the value at the vertex to -1 and at the point of the maximum eigenvalue to 1, ensuring the satisfaction of the limiting condition. We strive to minimize the components along the direction of the smaller eigenvalues, thus $r$ should ideally be as close to $a^{(n)}$ as possible. However, if $r$ is too small, it can lead to substantial growth in the direction of the larger eigenvalues. Through our calculations, we obtain $r = 0.146(a^{(1)} + a^{(n)}) \approx 0.1465a^{(1)}$ and $r = 0.8535(a^{(1)} + a^{(n)}) \approx 0.8535a^{(1)}$. the corresponding descending fuction is

$$h(x) = (1 - \frac{x}{0.8535a^{(1)}})(1 - \frac{x}{0.1465a^{(1)}}) = 7.998 * (0.8535 - y)(0.1465 - y) \tag{23}$$

where $y = \frac{x}{a^{(1)}}$,by combining $g(x)$ and $h(x)$, we obtain the final descending function $e(x) = g(x)h(x)$ After the above treatment, $e(x)$ have effectively removed the spikes of the eigenvectors outside the direction of the small eigenvalues, especially in the upper half-interval. This would make the current value of $r$ become smaller, with the larger components being very small. Therefore, even if we use the current smaller $r$ value twice in succession, there will not be a significant increase. Generally speaking, the primary effects are twofold: on the one hand, it eliminates the noise in the direction of the large eigenvalues; on the other hand, it shifts the value of r towards the direction of the small eigenvalues.

## 5 EXPERIMENT

Considering an example as follow

$$f(x) = \sum_{i=1}^{20000} a^{(i)}{x^{(i)}}^2 \tag{24}$$

, where the sequence $\{a^{(i)}\}$ is arithmetic progression and $0.1 \le a^{(i)} \le 10000$, $\{x_0^{(i)}\}$ is a random number between 0.2569 and 9999.6123.

We compared this algorithm with the RSD method, BB method, CBB method, SD method . For the EM method, we set the jump points $r_j$ to be 10, respectively.After the iterations, the minimum gradient norm values for the five methods are 6583.5315,0.0005,2.1152e-09,8.3415e-40 and 2.7352e-75, respectively.The results showed that the EM method is effcient for large scale problems. In second

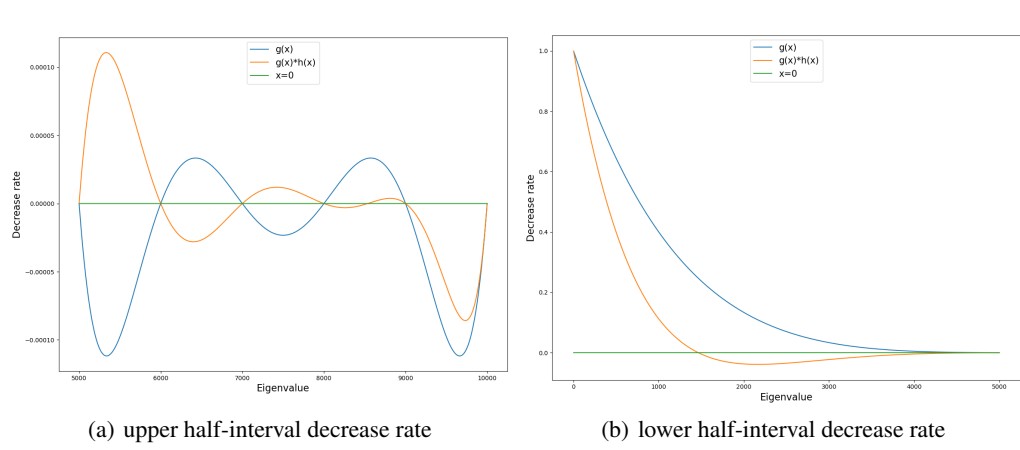

(a) upper half-interval decrease rate

(b) lower half-interval decrease rate

Figure 7: g(x) and h(x) decrease functions

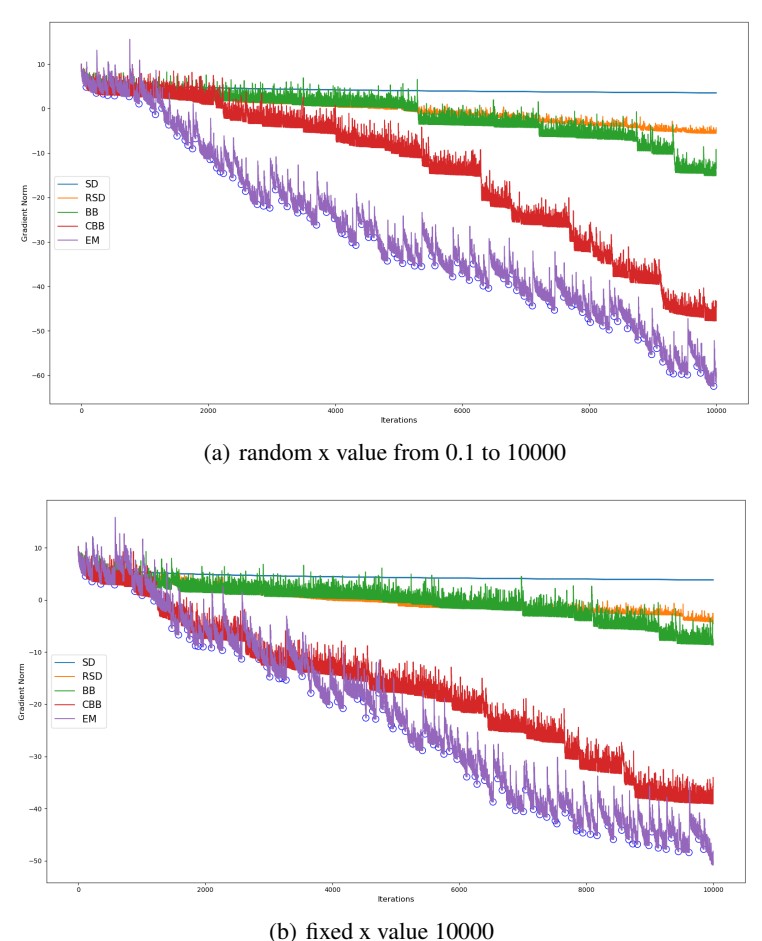

(a) random x value from 0.1 to 10000

(b) fixed x value 10000

Figure 8: Comparison of 5 methods

experiment we We choose the initial value of $\{x_0^{(i)}\}$ to be the same, which is 10000.The results are similar to the previous case, As shown in Figure(8), in the initial period, due to the large magnitude of each component, the EM method will experience a significant increase in the large characteristic components after a jump, which in turn leads to a significant increase in the overall gradient norm value.As the number of iterations continues to increase, both the BB and RSD methods will exhibit a relatively slow overall decline. The SD method will hardly change. After accumulating over a period of time, the CBB method shows a significant drop, which gives it an advantage compared to the aforementioned methods.while the EM method will show a significant downward trend after each jump.we report with circles the iteratios in which the current eigenvalue satisfies the condition.

## 6 CONCLUSION

When the current value of $r$ is at an extremely small eigenvalue, we achieve a significant decrease in the upper half of the eigenvalue spectrum and the maximum possible decrease in the lower half by selecting multiple different $r$ values within the intervals of large and small eigenvalues. In the future, we aim to improve the convergence speed by selecting more efficient descent functions.

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
