# OpenReview forum: "A New Efficient Method for Eliminating Minor Components"
_ICLR.cc/2026/Conference — Submitted to ICLR 2026_

### Official Review · Reviewer_LXF7 · 2025-10-26

**Soundness:** 3
**Presentation:** 2
**Contribution:** 3
**Rating:** 4
**Confidence:** 4

**Summary:**

The paper proposes an efficient method for eliminating minor component values in solving a quadratic programming problem. The theoretical analysis is presented, and two examples are studied to demonstrate the proposed method's efficiency.

**Strengths:**

1. The method can handle two and n-dimensional quadratic programming problems.
2. The theoretical analysis is discussed.
3. Two examples are studied, and the numerical results are provided.
4. A comparison of the EM method with the other four methods is made to show the EM method's efficiency.

**Weaknesses:**

1. See the paper's title. Please delete "CONFERENCE SUBMISSIONS".
2. Please make sure notations and symbols in the text are in italic.
3. Please follow the format to write citations in the text.
4. See page 2, the paragraph at the top. Please define those acronyms before using them.
5. See Figure 1. The caption is not suitable. Please revise it.
6. See page 3, line 125. Please use the proper comma (,) to separate those equations.
7. Please follow the format to prepare the references.
8. The writing of the paper is not clear.

**Questions:**

1. See the abstract. What are "BB and CBB"? Please define.
2. See the title for Section 2. What is "R ANALYSIS"? Please provide an easy-to-understand section title.
3. Please mention clearly the eigenvalue throughout the text. It is very confusing which symbol is used for eigenvalues, a or r?

---

### Official Review · Reviewer_TUdh · 2025-10-28

**Soundness:** 1
**Presentation:** 1
**Contribution:** 1
**Rating:** 0
**Confidence:** 5

**Summary:**

This paper proposes a new gradient-based method, called EM (Eliminating Minor Components), for solving quadratic optimization problems. The main idea is to eliminate the components in the direction of large eigenvalues when the current iterate lies in a direction dominated by small eigenvalues, in order to accelerate convergence.

**Strengths:**

The idea of eliminating components aligned with large eigenvalues in gradient-based quadratic optimization has some intuitive appeal. The work attempts to link its proposed EM method to well-known classical methods such as Steepest Descent (SD), Barzilai–Borwein (BB), and the Combined BB (CBB).

**Weaknesses:**

1. The paper is poorly presented and riddled with numerous writing, grammar, and spelling errors, with nearly every sentence containing a grammatical mistake. I find a significant part of the paper difficult to understand.
2. Several section titles are unclear and make the manuscript hard to read. For example, **R Analysis** and **N Dimensions Case**.
3. Lack of clear algorithm definition. The EM method is never formally stated.
4. Many equations and notations are presented without explanation. Variables such as "$r$", "$H(r)$", "$\mu(i)$", and "$W(i)$" are undefined or used inconsistently.
5. There are numerous formatting problems. For example, ordinary text within mathematical equations should be written in upright/roman font rather than italics.
6. Citations are incomplete and stylistically inconsistent.
7. Poor experimental design.
    (1) Synthetic test setup (a(i) between 0.1 and 10 000) lacks realism or justification.
    (2) No reproducible code or parameter details are given.
    (3) Results are reported as raw numbers without units or clear performance metrics.
    (4) Figures lack labeled axes or statistical meaning.

**Questions:**

See the **Weaknesses** section for details.

---

### Official Review · Reviewer_rBHv · 2025-10-31

**Soundness:** 2
**Presentation:** 1
**Contribution:** 1
**Rating:** 0
**Confidence:** 5

**Summary:**

This paper is proposing a gradient-based method for quadratic programming in the spirit of the classic Barzilai-Borwein scheme and its variants. The main idea is to control the contribution of large eigenvalues (of the matrix defining the problem) when the algorithm lies near the eigenspace for the smaller eigenvalues of the problem. The proposed scheme is doing this by doing a check on the value of the step-size based on the "spikes" of the eigenvectors present, and eliminating the minor components accordingly.

The paper does not provide any theorems, and only tests the proposed strategy in a toy example. It is a nice starting point for studying fine-grained step-size schemes for quadratic programming, and I encourage the authors to continue working on these ideas. However, the paper itself is not yet at a state where it can be considered for publication at ICLR.

My score of "0" should not be interpreted as a harsh criticism of the authors' work, but as my assessment of the distance that the paper would have to close in order to be considered for publication at a top-tier conference or journal.

**Strengths:**

The paper is a nice starting point and the geometric ideas are interesting.

**Weaknesses:**

The paper is not up to par in any of the three axes below:
- *Writing / Presentation:* The paper needs to be proof-read and the authors would need to provide much more context and explanations
- *Theoretical results:* The paper doesn't have any. While reasonable as an idea, the authors do not provide any theoretical results suggesting that the proposed method would improve on existing methods.
- *Numerical experiments:* For a paper without theoretical results, I would have expected a much more extensive numerical campaign than a single experiment.

Finally, there is a question of scope: quadratic programming is not at the core of the ICLR optimization community. If the authors could obtain results on the strongly convex / smooth regime, this would be more relevant for ICLR.

**Questions:**

None.

---

### Official Review · Reviewer_JkQ4 · 2025-11-01

**Soundness:** 1
**Presentation:** 1
**Contribution:** 1
**Rating:** 0
**Confidence:** 5

**Summary:**

The paper presents an analysis of the gradient descent method applied to quadratic functions. The writing and presentation are of poor quality and hard to follow. I would suggest the authors to revise the paper carefully before any submission.

**Strengths:**

Numerical experiments shows some improvements of the method.

**Weaknesses:**

Poor presentation. The contributions are very incremental.

**Questions:**

No

---

### Meta-Review · Area_Chair_WRwH · 2026-01-03

**Summary:**

This paper proposes a gradient descent method for quadratic objective functions, with a stepsize based on the classic Barzilai--Borwein method. Most reviewers note the poor quality of writing and contributions, and I agree with these concerns. In summary, this paper is not ready to submit in its current form, so I recommend **Reject**.

**Reviewer Concerns:**

This paper didn't submit rebuttals.

**Reviewer Scores:**

I think the reviewer would not change their score if they had been able to participate fully in the discussion.

---

### Decision · Program_Chairs · 2026-01-26

Reject